# REFLEX-MED: REINFORCEMENT WITH LABEL-FREE EXPLAINABILITY FOR UNIFIED MEDICAL REASONING

## ABSTRACT

Clinicians urgently need explanations they can audit, not merely fluent chains. Yet prevailing practices conflate interpretability with subjective human/LLM rationales, with post-hoc visuals loosely aligned to answers, or with answer rationale consistency. These proxies are annotation-hungry, bias-prone, and crucially do not certify *process verifiability*: where the model looked and why it looked there. Meanwhile, reinforcement learning from feedback excels at answer verifiability but offers little support for constraining the provenance of attention or penalizing visually ungrounded reasoning. We introduce **REFLEX-Med**, a reinforcement framework that instantiates label-free explainability through two verifiable prerequisites: *(i) faithful visual grounding* that is text-conditioned localization in the image, and *(ii) bi-directional cross-modal provenance*, that is a cycle of mutual traceability across image-text and text-text semantics. REFLEX-Med couples curriculum GRPO with two frozen rewards computed by a medical vision-language encoder: a visual fidelity reward aligning text-conditioned saliency between the model's own answer and an anchor text, and a bi-modal provenance reward enforcing image-text and text-text consistency in embedding space. Together with standard format and semantic-matching rewards, REFLEX-Med resists large VLM hallucination and attention-think drift, improving both answer quality and auditable faithfulness on unified medical reasoning (open and close-ended VQA) all without human or LLM rationale annotations.

## 1 INTRODUCTION

Medical AI systems (Moor et al., 2023; Wu et al., 2023; Huang et al., 2025) are increasingly judged not only by *what* they answer but also by *where* they attend and *why*. In high stakes clinical workflows, this demand for auditable explanations is urgent: clinicians must be able to verify that a system's conclusions are grounded in appropriate image evidence and clinically sensible semantics. Yet prevailing practices conflate interpretability with subjective artifacts. Human-authored or LLM-generated rationales are annotation-hungry (Xu et al., 2024) and bias-prone (Wang et al., 2024); post-hoc heatmaps (Stan et al., 2024) often bear a tenuous relation to the decision pathway; and "answer-rationale consistency" collapses explanation to a by-product of correctness (Zhang et al., 2024b). None of these reliably certifies *process verifiability*, the capacity to check, without bespoke labels, where a model looked and why it looked there.

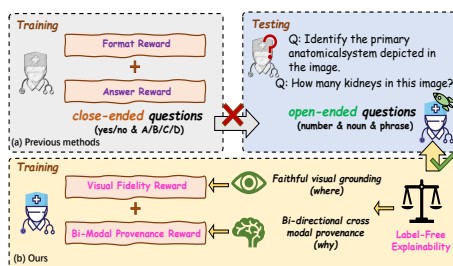

Figure 1: (a) Rigid rewards cause models to fail at tasks requiring high freedom. (b) We introduce label-free explainability guides models toward human-like perception, offering flexibility and scalability.

Reinforcement learning from human feedback (RLHF) (Kaufmann et al., 2024) has recently delivered strong gains on answer verifiability by optimizing rule-verifiable objectives (DeepSeek-AI, 2025). However, current RL pipelines provide little support for explanation verifiability: they rarely constrain the *provenance* of attention (Miao et al., 2024), nor do they penalize semantically plausible yet visually ungrounded reasoning (answer-right, look wrong). These LVLMs are vulnerable to hallucination (Wu et al., 2025b) and what we term attention-think drift: fluent chains of thought unmoored from the visual evidence (Shao et al., 2024).

We address this gap with **REFLEX-Med** (Reinforcement for Label-Free Explainability in Unified Medical Reasoning), a reinforcement framework that makes explanation *auditable* and *label free*. Our premise is to ground explainability in two verifiable prerequisites:

1. **Faithful visual grounding** (*where*): text-conditioned localization in the image should align with the evidence implicated by the answer.

2. **Bi-directional cross-modal provenance** (*why*): image-text and text-text semantics must close a mutual traceability loop, so that the answer's language and an anchor description point to each other and to the same image evidence.

Operationally, REFLEX-Med couples curriculum reinforcement learning with auditable judges. We compute two label-free[1] rewards: (i) a *Visual Fidelity Reward* aligning text-conditioned saliency between the model's answer and anchor text via IoU of binarized saliency maps; (ii) a *Bi-modal Provenance Reward* enforcing high-level embedding-space agreement marginalized for stability. These explanation rewards complement standard format and semantic matching rewards in medical VQA, and crucially, the frozen judges prevent reward hacking and non-stationarity without human or LLM rationale labels. Our core goal is to *verify if injecting human-like perceptual label-free explainability can make LVLMs more like human experts*.

To align with clinical practice, we cast the task as Unified Medical Reasoning, one policy handling both close-ended and open-ended VQA. By optimizing process verifiability directly, REFLEX-Med must not only produce correct answers but also ground them and close the provenance loop across modalities. Our contributions are summarized as follows:

- We put forward a label-free and auditable notion of explainability for medical reasoning based on verifiable prerequisites: *faithful visual grounding* and *bi-directional cross-modal provenance*, re-framing explanation from subjective rationales to evidence that can be checked without new labels.

- We introduce **REFLEX-Med**, which integrates two explanation rewards, visual fidelity and bi-modal provenance reward, into curriculum reinforcement learning, thereby optimizing *process verifiability* alongside answer quality.

- We show that using frozen medical vision language encoders as immutable judges *resists* large LVLM hallucination and attention-think drift, avoiding non-stationarity and reward hacking while requiring no human/LLM rationale annotations.

## 2 RELATED WORK

**Reinforcement Learning in LLMs/LVLMs**. Recent work has scaled outcome-driven RL beyond supervised instruction tuning for text-only LLMs and vision-language models. Group-relative policy optimization (GRPO) (DeepSeek-AI, 2025) emerged as a value-free PPO variant and has boosted symbolic and mathematical reasoning. Subsequent analyses (Du et al., 2025; Wang et al., 2025; Huang et al., 2025) and variants explore off-policy estimation and theoretical properties, positioning GRPO as a practical backbone for reasoning-oriented RL fine-tuning. In parallel, industry-scale efforts (Team, 2025) show RL can close

---

[1]Throughout, "label-free" means we do not use human-annotated rationales (e.g., boxes/masks or chains of thought) nor LLM-generated rationales for supervision.

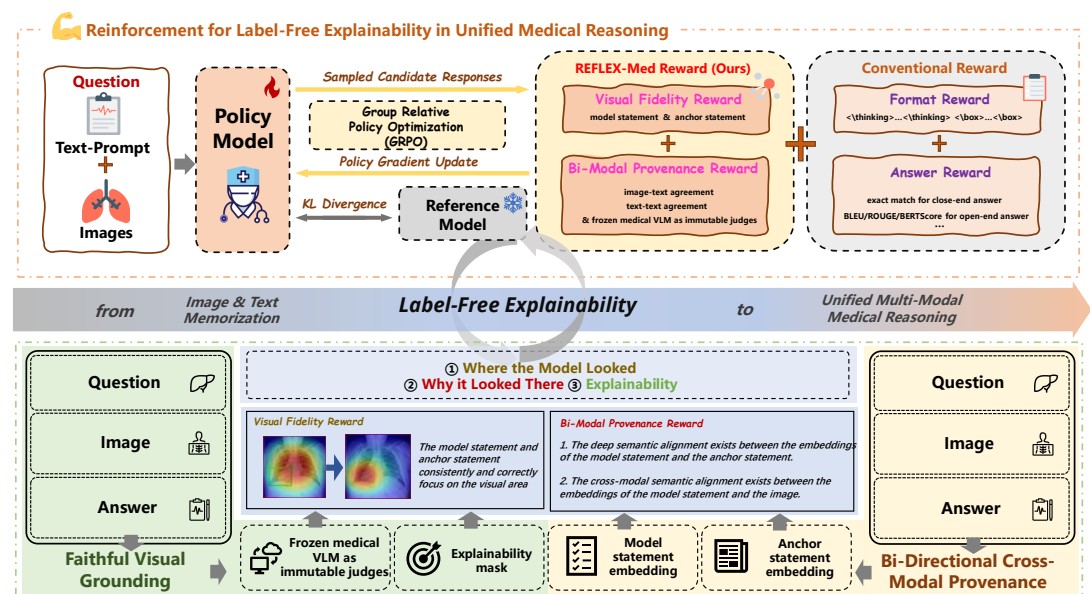

Figure 2: Overview of REFLEX-Med. Visual Fidelity Reward (IoU between explainability masks from the model and anchor statements) and Bi-modal Provenance Reward (agreement of image-to-text and text-to-text embeddings), combined with format and answer rewards, yield unified medical reasoning.

much of the gap between smaller models and very large proprietary reasoners. Beyond outcome-only rewards, preference data alignment continues to evolve. RLAIF-V (Yu et al., 2025) adapts AI feedback-based RL to large hallucination reductions. Our method follows this post-instruction alignment wave but differs in using rewards focused on explanation verifiability not just answer correctness.

**Multimodal Medical Reasoning**. Medical LVLMs have advanced from domain-adapted instruction tuning to broader foundation modeling, but most optimize answer utility (Lai et al., 2025; Pan et al., 2025) without auditable visual grounding or process verifiability constraints. Early systems like LLaVA-series (Liu et al., 2024; Li et al., 2023) and Med-Flamingo (Moor et al., 2023) adapted general VLMs to biomedical images via large-scale instruction data and few-shot workflows. Recent medical work adds larger corpora, architectures (Chen et al., 2024; Zhang et al., 2025), and evaluations covering multi-image reasoning and 3D modalities. Very recent work (Pan et al., 2025; Wu et al., 2025a) uses GRPO-style reinforcement fine-tuning for medical reasoning, reporting in-domain/out-of-domain gains. Our study follows this paradigm but focuses on *process verifiability* in a unified setting evaluating both close- and open-ended medical reasoning.

## 3 METHODOLOGY

We implement **REFLEX-Med** with curriculum reinforcement learning. The method is guided by two auditable desiderata, faithful visual grounding (where) and Bi-directional cross-modal provenance (why), to make explanations label-free and process verifiable. Concretely, we *(i)* convert heterogeneous VQA formats into canonical statements, *(ii)* obtain explainability masks and semantic embeddings from a frozen mmedical vision-language encoder, *(iii)* optimize two explanation rewards, Visual Fidelity Reward and Bi-modal Provenance Reward, and *(iv)* train from close-ended to open-ended, enabling unified medical reasoning.

## 3.1 UNIFIED INTERFACE AND FROZEN JUDGES

To evaluate explanations in a format-agnostic, auditable way, we *declarativize* each question–answer into a canonical statement with $\text{decl}(\cdot, \cdot)$. Let $I$ be the image, $q$ the question, and $a$ the policy's answer from $\pi_\theta$. The deterministic operator maps polar, multi-choice, and open-ended prompts to concise factual assertions for the frozen judge to assess localization and semantics. The anchor uses the dataset answer $a^\star$, i.e., $t_2 = \text{decl}(q, a^\star)$, requiring no rationale labels and preserving our label-free, process-verifiable design.

$$\underbrace{t_1}_{\text{model statement}} = \text{decl}(q, a), \qquad \underbrace{t_2}_{\text{anchor statement}} = \text{decl}(q, a^\star), \qquad \text{decl}: Q \times \mathcal{A} \to \mathcal{T}. \qquad (1)$$

The frozen medical vision-language encoder ($\phi_{\text{img}}, \phi_{\text{text}}$) provides (i) unit-norm image and text embeddings capturing high-level medical semantics, and (ii) a text-conditioned saliency map indicating where a statement attends in the image. We binarize saliency with a quantile threshold to fix coverage and make masks comparable across images and statements, which is crucial for auditable, label-free localization

$$v = \phi_{\text{img}}(I) \in \mathbb{R}^d, \quad u_k = \phi_{\text{text}}(t_k) \in \mathbb{R}^d \ (k \in \{1, 2\}), \quad \|v\|_2 = \|u_k\|_2 = 1, \quad S(I, t) \in [0, 1]^{H \times W}. \qquad (2)$$

Specifically, the image embedding $v$ and the text embeddings $u_k$ (for $t_1$ and $t_2$) are $L_2$-normalized. The text-conditioned saliency is $S(I, t) \in [0, 1]^{H \times W}$. We obtain a discrete explainability mask by thresholding at the $p$-quantile, as formalized below:

$$\tau_p(I, t) = \text{Quant}_p(S(I, t)), \quad M(I, t) = \mathbb{I}\big[S(I, t) \geq \tau_p(I, t)\big], \qquad (3)$$

where $\mathbb{I}[\cdot]$ is indicator the function and $\text{Quant}_p$ returns the $p$-th percentile and $p{=}0.5$ by default. The tuple $\big(u_k, M(I, t_k)\big)$ is the auditable evidence for *why* and *where*.

## 3.2 VISUAL FIDELITY REWARD (VFR): FAITHFUL VISUAL GROUNDING

Clinical users must verify where a conclusion comes from. To encode faithful visual grounding, we require the model's focus to spatially coincide with the anchor's focus on $I$. Using the explainability masks $M_1$ and $M_2$ from Equation 3 (Figure 3 show the logic for determining whether a model should be rewarded), we compute their intersection-over-union (IoU), which quantifies the proportion of shared visual evidence relative to the joint support; this directly operationalizes the "answer-right, look-right" criterion that process verifiability demands:

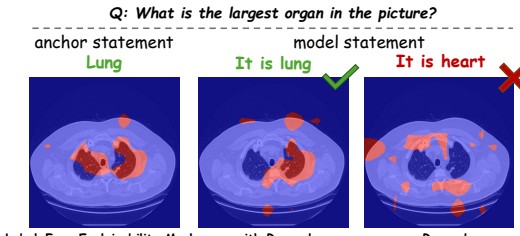

Figure 3: Mask $M_{1/2}$ (red) and whether rewarded.

$$\text{IoU}(M_1, M_2) = \frac{|M_1 \cap M_2|}{|M_1 \cup M_2|} \in [0, 1], \qquad M_1 = M(I, t_1), \ M_2 = M(I, t_2). \qquad (4)$$

The *Visual Fidelity Reward* grants a discrete bonus when spatial agreement exceeds a clinically meaningful margin (default $\tau_{\text{IoU}}{=}0.5$). Discreteness stabilizes group-standardized advantages in GRPO:

$$R_{\text{VFR}} = \mathbf{1}\Big[ \text{IoU}(M_1, M_2) > \tau_{\text{IoU}} \Big], \qquad \tau_{\text{IoU}} = 0.5. \tag{5}$$

Intuitively, Equations 4 and 5 reward policies that "look where the evidence lives" and penalize visually ungrounded shortcuts. This directly targets the introduction's failure mode (answer-right, look-wrong) and makes the *where*-axis of process verifiability auditable without human/LLM rationale supervision.

### 3.3 BI-MODAL PROVENANCE REWARD (BPR): BI-DIRECTIONAL CROSS-MODAL PROVENANCE

Auditable explanations also require *why* the statement is semantically supported by both the image and the anchor language. We enforce bi-directional cross-modal provenance as a two-link loop: (i) text–text agreement between the model statement and the anchor, and (ii) image–text agreement between the model statement and the image. Each link is measured by cosine similarity in the frozen vision and language encoder space, providing label-free yet clinically grounded semantics:

$$s_{\text{tt}} = \langle u_1, u_2 \rangle, \qquad s_{\text{it}} = \langle v, u_1 \rangle, \qquad u_1 = \phi_{\text{text}}(t_1), \ u_2 = \phi_{\text{text}}(t_2), \ v = \phi_{\text{img}}(I). \tag{6}$$

We require each similarity to exceed a margin that captures language specificity and imaging variability. BPR awards two half-credits when these thresholds are met, discouraging fluent but ungrounded statements and mitigating the attention–think drift highlighted in the introduction.

$$R_{\text{BPR}} = \frac{1}{2}\,\mathbb{I}[s_{\text{tt}} > \tau_{\text{tt}}] + \frac{1}{2}\,\mathbb{I}[s_{\text{it}} > \tau_{\text{it}}], \qquad \tau_{\text{tt}} = 0.8, \ \tau_{\text{it}} = 0.5. \tag{7}$$

We summarize the loop tightness by a marginized score that vanishes unless both links are satisfied; it mirrors the intuition that provenance should *close* across modalities and language, strengthening the definition of process verifiability without introducing any new supervision channels or proxy labels:

$$\text{LoopTight}(I, q, a) = \left(\min\big\{\, s_{\text{tt}} - \tau_{\text{tt}}, \ s_{\text{it}} - \tau_{\text{it}} \,\big\}\right)_+, \qquad (\cdot)_+ = \max(0, \cdot). \tag{8}$$

Together, Equations 6 to 8 realize the *why*-axis of process verifiability: the model's claim must be semantically traceable to both the anchor and the image via a frozen, auditable judge. This resists LVLM hallucination by making it costly to produce language that is decoupled from clinical evidence.

### 3.4 REINFORCEMENT LEARNING WITH CURRICULUM FOR UNIFIED MEDICAL REASONING

We integrate explanation and utility under reinforcement learning, following recent GRPO-style fine-tuning while preserving our label-free, auditable design. Alongside $R_{\text{VFR}}$ and $R_{\text{BPR}}$, we include a format reward $R_{\text{fmt}} \in \{0, 1\}$ for the `<think>`/`<answer>` schema and an answer reward $R_{\text{ans}}$ (exact match for close-ended and BLEU/ROUGE/BERTScore for open-ended) ensuring that the agent remains clinically useful rather than over-optimized for explanations alone.

$$R_{\textit{REFLEX-Med}} = R_{\text{ans}} + R_{\text{fmt}} + R_{\text{VFR}} + R_{\text{BPR}}. \tag{9}$$

For each prompt $(I, q)$, we sample a group $\{y_i = (c_i, a_i)\}_{i=1}^{G} \sim \pi_\theta(\cdot \mid I, q)$, compute $\{R_i\}$ via Equation 9, standardize within-group to $A_i = (R_i - \mu)/\sigma$, and update using a clipped, reference-conditioned objective with KL regularization, stabilizing exploration and mitigating degenerate drift while the frozen judge keeps rewards stationary and auditable

Table 1: Performance of our REFLEX-Med and different type of VLM on three in/out-of-domain datasets. c.: close-end accuracy; o.: open-ended metrics; R: reasoning; U: understanding.

| Model | In-domain test | | | | | | | Out-of-domain test | | | | | |
| | VQA-RAD | | SLAKE | | Path-VQA | | Avg. | Quilt-VQA | | PMC | MedXpert | | Avg |
| | c. | o. | c. | o. | c. | o. | | c. | o. | c. | R.c. | U.c. | |
| *General VLM* | | | | | | | | | | | | | |
| Yi-VL-34B | 53.0 | 22.4 | 58.9 | 33.9 | 47.3 | 12.9 | 38.1 | 56.0 | 13.2 | 39.5 | 19.9 | 20.7 | 29.9 |
| LLaVA-v1.6-7B | 52.6 | 19.8 | 57.9 | 37.6 | 47.9 | 12.6 | 38.1 | 58.3 | 8.7 | 35.5 | 20.7 | 20.6 | 28.8 |
| LLaVA-v1.6-13B | 55.8 | 24.0 | 58.9 | 44.5 | 51.9 | 12.8 | 41.3 | 57.4 | 24.5 | 36.6 | 19.5 | 18.1 | 31.2 |
| LLaVA-v1.6-34B | 58.6 | 24.1 | 67.3 | 44.6 | 59.1 | 15.0 | 44.8 | 62.4 | 23.7 | 44.4 | 20.6 | **25.5** | 35.3 |
| Qwen2.5-VL-7B | 67.3 | 32.2 | 71.6 | 40.2 | 65.5 | 17.2 | 49.0 | 54.8 | 29.0 | 50.4 | 20.6 | 23.1 | 35.6 |
| *Finetuned VLM* | | | | | | | | | | | | | |
| Qwen2.5-VL-7B (SFT) | 71.3 | 27.8 | 78.6 | 50.8 | **87.8** | **33.6** | 58.3 | 60.9 | 8.9 | 49.2 | 20.2 | 20.4 | 31.9 |
| Qwen2.5-VL-7B (GRPO) | 70.5 | 29.8 | 79.3 | 40.2 | 82.8 | 27.8 | 55.1 | 50.2 | 28.4 | 51.2 | 21.2 | 21.7 | 34.5 |
| *Medical VLM* | | | | | | | | | | | | | |
| Med-Flamingo-7B | 45.4 | 29.3 | 43.5 | 30.1 | 54.7 | 28.7 | 38.6 | 62.1 | 22.3 | 23.3 | 19.0 | 20.0 | 29.3 |
| RadFM-13B | 50.6 | 34.0 | 34.6 | 44.2 | 38.7 | 19.9 | 37.0 | 60.7 | 21.5 | 25.9 | 19.8 | 19.6 | 29.5 |
| LLaVA-Med-7B | 51.4 | 10.1 | 48.6 | 6.6 | 56.8 | 8.4 | 30.3 | 63.0 | 29.3 | 24.7 | 20.5 | 19.5 | 31.4 |
| HuatuoGPT-Vision-8B | 63.8 | 36.0 | 74.5 | 47.0 | 59.9 | 23.2 | 50.7 | 63.9 | 38.5 | 52.7 | 20.4 | 22.9 | 39.6 |
| REFLEX-Med-3B | 72.3 | 35.6 | 80.1 | 61.8 | 83.0 | 26.8 | 60.0 | 64.2 | 37.1 | 54.0 | 22.8 | 22.1 | 40.4 |
| REFLEX-Med-7B | **78.2** | **41.1** | **80.9** | **66.3** | 84.0 | 30.3 | **63.4** | **70.2** | **40.0** | **54.5** | **24.0** | 23.9 | **42.5** |

$$\mathcal{L}_{\text{GRPO}} = \frac{1}{G} \sum_{i=1}^{G} \min\Big( r_i A_i,\ \text{clip}(r_i, 1-\epsilon, 1+\epsilon)\, A_i \Big) - \beta\, \text{KL}\Big( \pi_\theta(\cdot \mid I, q) \,\big\|\, \pi_{\text{ref}}(\cdot \mid I, q) \Big), \qquad r_i = \frac{\pi_\theta(y_i \mid I, q)}{\pi_{\text{ref}}(y_i \mid I, q)}. \tag{10}$$

Training uses a two-phase curriculum for unified medical reasoning. Phase I (close-ended) uses low-variance exact-match $R_{\text{ans}}$ to establish reliable decision boundaries. Phase II (open-ended) introduces semantic $R_{\text{ans}}$ (BLEU/ROUGE/BERTScore). This schedule reduces variance and gradient conflict between discrete and continuous signals while continuously rewarding faithful visual grounding and bi-directional cross-modal provenance, addressing the process verifiability requirement in the introduction.

## 4 EXPERIMENTS

**Datasets and Benchmarks.** To ensure direct comparability, we follow the data recipe of previous works (Rui et al., 2025; Chen et al., 2024). The training pool is the union of VQA-RAD (Lau et al., 2018), SLAKE (Liu et al., 2021), and PathVQA (He et al., 2020), totaling $\sim$ 27k close- and open-ended QA pairs. We use the official splits and do not add rationale labels or external supervision. For evaluation we use in-domain test sets (VQA-RAD, SLAKE, PathVQA) and out-of-domain suites including PMC-VQA (Zhang et al., 2023), Quilt-VQA (Seyfioglu et al., 2024), the multimodal MedXpertQA subset (Zuo et al., 2025),

and the MMMU Health & Medicine track (Yue et al., 2024). This setup probes answer quality and the two explanation desiderata, faithful visual grounding and bi-directional cross-modal provenance.

**Baselines.** We compare against general vision-language models such as Yi-VL (Young et al., 2024), LLaVA-v1.6 (Liu et al., 2024) with multiple sizes, and Qwen2.5-VL (Bai et al., 2025), medical vision-language models such as Med-Flamingo (Moor et al., 2023), RadFM (Wu et al., 2023), LLaVA-Med (Li et al., 2023), and HuatuoGPT-Vision (Chen et al., 2024), and backbone-controlled finetunes on Qwen2.5-VL (Bai et al., 2025) including supervised fine-tuning and vanilla GRPO. All baselines use the same data, splits, and metrics. Our method is REFLEX-Med, which augments GRPO with the Visual Fidelity Reward and the Bi-modal Provenance Reward while keeping the medical judge frozen and label-free.

**Implementation Details.** The backbone is Qwen2.5-VL (Bai et al., 2025)-7B-Instruct, with a 3B variant used for scale ablations. Training is in PyTorch using verl (Sheng et al., 2024) for GRPO and vLLM (Kwon et al., 2023) on NVIDIA RTX 5880×8. The medical judge is a BioMedCLIP (Zhang et al., 2024a) kept frozen. GRPO settings: group size 8, temperature 1.0, batch 64, learning rate $1 \times 10^{-6}$, reference KL weight $\beta = 0.01$. Outputs follow the `<think>` / `<answer>` schema. The curriculum runs close to open, one epoch per phase under the same RL budget as previous work (Rui et al., 2025).

## 4.1 MAIN RESULTS

**In-Domain and Out-of-Domain Generalization.** REFLEX-Med-7B achieves the best averages in Table 1, with an in-domain average of 63.4% and an out-of-domain average of 42.0%. Compared with standard GRPO on the same backbone, the in-domain score rises from 55.1% to 63.4%, while the out-of-domain score increases from 34.5% to 42.5%, indicating reduced hallucination and more stable reasoning. The model also excels on open-format reasoning, reaching 66.3% on SLAKE open questions, which supports genuine open-world inference rather than memorization. These gains stem from declarativized statements judged by a frozen medical encoder together with the Visual Fidelity Reward and the Bi-modal Provenance Reward, which align attention to image evidence and enforce semantic traceability.

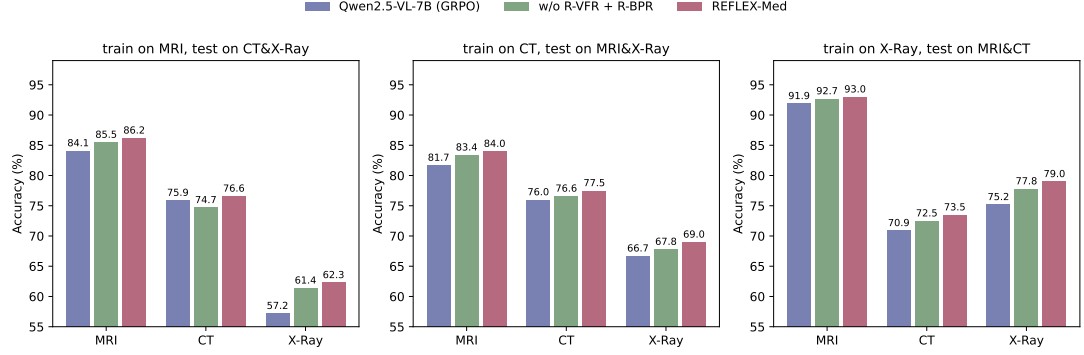

Figure 4: Cross-modal performance on SLAKE (Liu et al., 2021), where each model is trained on a single modality and evaluated across all modalities to enable in-modal and cross-modal comparison.

**Cross-Modal Generalization.** We train on one modality and test on the other two to probe transfer across MRI, CT, and X-ray. REFLEX-Med-7B is consistently the top curve: training on MRI yields 62.3% on X-ray, training on CT yields 69.0% on X-ray, and training on X-ray yields 73.5% on CT. The ablation without the VFR and the BPR falls between vanilla GRPO and REFLEX-Med, showing that faithful visual

Table 2: Zero-shot generalization for the selected Health&Medicine track of MMMU (Yue et al., 2024), covering category-wise and overall accuracy. The abbreviations are defined as: BMS (Basic Medical Science), CM (Clinical Medicine), DLM (Diagnostics and Laboratory Medicine), P (Pharmacy), PH (Public Health).

| Model | BMS | CM | DLM | P | PH | MMMU Health & Medicine |
|---|---|---|---|---|---|---|
| *General VLM* | | | | | | |
| Yi-VL-34B | 49.4 | 48.9 | 43.2 | 40.5 | 32.0 | 41.5 |
| LLaVA-v1.6-7B | 40.5 | 36.9 | 32.1 | 32.3 | 26.9 | 33.1 |
| LLaVA-v1.6-13B | 53.6 | 46.7 | 33.3 | 22.2 | 40.0 | 39.3 |
| LLaVA-v1.6-34B | 56.4 | 56.0 | 46.9 | 46.7 | 41.7 | 48.8 |
| LLaVA-v1.5-LLaMA3-8B | 42.3 | 44.0 | 37.0 | 34.7 | 35.2 | 38.2 |
| Qwen2.5-VL-7B | 50.0 | 63.3 | 33.3 | 59.3 | 53.3 | 51.7 |
| *Finetuned VLM* | | | | | | |
| Qwen2.5-VL-7B (SFT) | 46.4 | 46.7 | 40.0 | 55.6 | 50.0 | 51.7 |
| Qwen2.5-VL-7B (GRPO) | 57.1 | 66.7 | 30.0 | 70.4 | 63.3 | 57.2 |
| *Medical VLM* | | | | | | |
| Med-Flamingo | 29.6 | 28.1 | 24.8 | 25.3 | 31.2 | 28.3 |
| RadFM | 27.5 | 26.8 | 25.8 | 24.7 | 29.1 | 27.0 |
| LLaVA-Med-7B | 39.9 | 39.1 | 34.6 | 37.4 | 34.0 | 36.9 |
| HuatuoGPT-Vision-8B | **61.0** | 58.8 | **50.0** | 44.7 | 38.7 | 49.1 |
| REFLEX-Med-7B | 57.3 | **67.0** | 40.3 | **73.5** | **67.6** | **61.1** |

grounding and bi-directional cross-modal provenance additively enhance transfer. Declarativized statements judged by a frozen medical encoder provide a label-free, modality-agnostic signal, aligning attention with evidence and enforcing semantic traceability. These results indicate that the policy acquires transferable, process-verifiable reasoning rather than memorizing modality-specific patterns.

**Zero-Shot Generalization.** As shown in Table 2, on the Health & Medicine track, REFLEX-Med-7B achieves strong zero-shot transfer: Clinical Medicine 67.0%, Pharmacy 73.5%, and Diagnostics and Laboratory Medicine 40.3%. Diagnostics improves by 10.3% versus the GRPO baseline, showing substantial gain on a hard subtask. These results mirror our in- and out-of-domain findings and suggest that declarativized statements judged by a frozen medical encoder, together with the VFR and the BPR, yield label-free, process-verifiable shaping that improves open-world medical reasoning.

**Reasoning Quality.** Figure 5 illustrates typical outputs under our structured format with `<think>` and `<answer>`. In the open-ended case (a), the chain explicitly performs comparative localization ("more prominent on the patient's left side") and ties the conclusion to the larger, more defined structure. In the closed-ended case (b), the chain justifies a negative decision by referencing relevant anatomy (temporal region, sphenoid sinus, skull base) and the absence of disruption, rather than resorting to generic negations. Qualitatively, the chains are concise, evidence-citing, supporting our claim that REFLEX-Med optimizes *pro-*

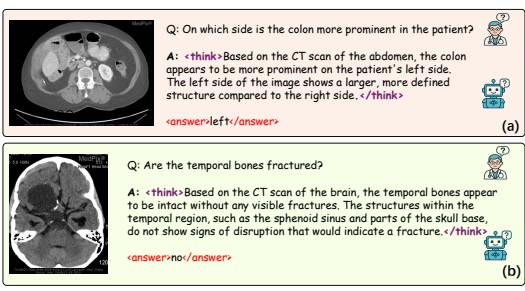

Figure 5: Qualitative results of (a) open-ended and (b) close-ended VQA reasoning.

*cess verifiability* and produces grounded reasoning in-
stead of template-like or speculative narratives.

## 4.2 ABLATION STUDY AND ANALYSIS

**Ablation on Model Size and Components.** As shown in Table 1, REFLEX-Med-3B achieves 60.0% in-domain and 40.4% out-of-domain, while HuatuoGPT-Vision-8B reports 50.7% and 39.6%, indicating that our label-free, process-verifiable shaping is effective even at small scale. Removing the key components (w/o $R_{VFR}$ and $R_{BPR}$) weakens cross-modal transfer in Figure 4; for example, CT→X-ray records 67.8% under ablation and 69.0% with the full model, and X-ray→CT records 72.5% under ablation and 73.5% with the full model. Across panels, the ablated variant typically lies between vanilla GRPO and REFLEX-Med, underscoring the contribution of grounding and provenance constraints.

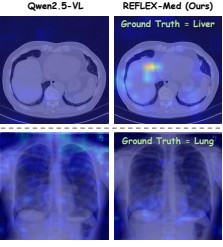

Figure 6: Attention.

**Analysis on Internal Attention.** Figure 6 contrasts internal attention maps from the vanilla Qwen2.5-VL and REFLEX-Med on CT and chest X-ray. The baseline exhibits diffuse, scattered activations over background structures, a signature of attention–think drift. In contrast, REFLEX-Med produces compact, high-confidence foci that coincide with the clinically implicated region in each case, yielding visibly higher overlap with the anchor's focus and fewer spurious hotspots. We attribute this behavior to our label-free, process-verifiable shaping. Qualitatively, the maps are sharper and sparser, indicating lower attention entropy and stronger localization faithfulness. These observations support our claim that REFLEX-Med grounds its answers in image evidence rather than relying on fluent but unmoored chains, tightening the provenance loop central to our motivation.

**Analysis on Dynamics of $R_{VFR}$ and $R_{BPR}$.** Figure 7 shows three reward traces. All three follow a similar pattern: rapid gains in early training followed by slower, steady increases. The concurrent upward trends indicate these signals are complementary rather than competing. Each curve approaches its empirical maximum, suggesting the model increasingly depends on semantic and localization cues and produces more correct, grounded outputs. This steady progression of rewards also aligns with the model's growing ability to meet the explainability requirements for medical VQA tasks. The lack of divergence or collapse indicates no obvious reward hacking, supporting the stability and effectiveness of the label-free explainability reward design.

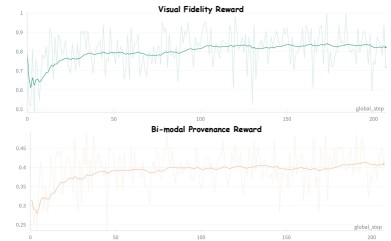

Figure 7: Reward dynamics.

## 5 CONCLUSION

We presented REFLEX-Med, a reinforcement framework that makes explanation auditable and label-free for unified medical reasoning; our core goal with this framework is to verify if injecting human-like perceptual label-free explainability can make LVLMs more like human experts. Instead of supervising text rationales or post-hoc heatmaps, REFLEX-Med optimizes two verifiable rewards: a Visual Fidelity Reward that aligns model saliency with text anchors, and a Bi-modal Provenance Reward that enforces mutual traceability between image–text and text–text semantics. With curriculum reinforcement learning and frozen medical VLM judges, the training signal remains stationary and resists attention–think drift. Across in-domain and out-of-domain evaluations, REFLEX-Med improves answer utility and explanation faithfulness, with strong zero-shot transfer across MRI/CT/X-ray and health benchmark; ablations show both rewards are necessary. By optimizing verifiable criteria rather than labels, the method narrows the gap between answer accuracy and explanation reliability and slots into existing RLHF pipelines with minimal changes.

## ETHICS STATEMENT

This study strictly adheres to the ethical guidelines and submission requirements of ICLR. The data and code used are legally sourced, with no unauthorized usage. The experimental code is either independently developed or reasonably modified based on open-source projects, in compliance with intellectual property regulations, and is prepared for public release as required. All authors declare no relevant conflicts of interest, and the research conclusions are not unduly influenced. The entire work fully meets the academic ethics and compliance standards of ICLR.

## REPRODUCIBILITY STATEMENT

The code is provided in the supplementary materials to replicate the empirical results.

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

# A APPENDIX

**Use Of LLMs.** We use large language models solely for language polishing of the final manuscript—correcting grammatical errors and refining expression. The models play no part in conceptualization, experimental design, theoretical analysis, or any substantive writing. All scientific viewpoints and results remain our sole responsibility.

