# OpenReview forum: "REFLEX-Med: Reinforcement for Label-Free Explainability  in Unified Medical Reasoning"
_ICLR.cc/2026/Conference — Submitted to ICLR 2026_

### Official Review · Reviewer_JQbZ · 2025-10-27

**Soundness:** 3
**Presentation:** 2
**Contribution:** 2
**Rating:** 4
**Confidence:** 4

**Summary:**

This paper introduces REFLEX-Med, a RL framework designed to improve the explainability of medical VLM without relying on costly human-annotated rationales. The core idea is to instantiate "label-free explainability" through two verifiable prerequisites: faithful visual grounding (where the model looks) and bi-directional cross-modal provenance (why it looks there).

To achieve this, the authors propose two novel reward signals computed by a frozen, pre-trained medical VLM acting as a "judge":
1.  A **Visual Fidelity Reward ($R_{VFR}$)**, which encourages the policy's attention (saliency map) to align with the attention of an "anchor" (ground-truth) statement.
2.  A **Bi-modal Provenance Reward ($R_{BPR}$)**, which enforces semantic consistency in the embedding space between the model's generated answer, the anchor text, and the input image.

**Strengths:**

1.  **Originality & Significance:** The paper proposes a novel and important shift in medical VLM explainability, moving from subjective (and annotation-hungry) rationales to objective, "label-free" verifiable criteria. The core concept of using a frozen judge to reward faithful grounding ($R_{VFR}$) and semantic provenance ($R_{BPR}$) is a creative and promising approach.
2.  **Problem Formulation:** The work correctly identifies a critical failure mode of current VLMs ("attention-think drift") and proposes a concrete mechanism to penalize it. The goal of instantiating "process verifiability" (where and why the model looked) is highly relevant for high-stakes domains like medicine.
3.  **Methodology:** The design of the two reward functions is intuitive and directly maps to the stated goals. Using a frozen judge to provide a stationary reward signal and prevent reward hacking is a sound design choice within an RL framework.

**Weaknesses:**

1.  **Unexplained Catastrophic Performance on PathVQA:** The most glaring weakness is the model's performance on the PathVQA dataset, as shown in Table 1. The Qwen2.5-VL (SFT) baseline achieves 87.8% (c) and 79.3% (o). In contrast, REFLEX-Med-7B scores 80.9% (closed) and a shockingly low 30.3% (o). This is a massive performance degradation on an in-domain dataset. The paper fails to acknowledge, analyze, or explain this result. This strongly suggests that the proposed reward framework may be fundamentally flawed or, at best, highly detrimental to specific modalities like pathology. This single result undermines the paper's primary claims of improving answer quality.
2.  **Unvalidated Reward Signal Quality:** The methodology critically depends on the frozen BioMedCLIP judge providing accurate saliency maps and meaningful embeddings. The paper provides zero evidence that BioMedCLIP is a reliable judge, especially for saliency. Medical grounding models are notoriously unreliable outside of the domain they were trained on (e.g., CXR). If the judge produces low-quality masks for CT or pathology images, the $R_{VFR}$ signal is optimizing the policy for noise, which would explain the poor performance on PathVQA. The authors must validate the judge's performance before using it as a source of truth.
3.  **Outdated and Limited Experimental Setup:**
    * **Baselines:** The baselines are missing more recent, state-of-the-art VLM, such as those from the InternVL series or the newer Qwen-VL models and InternVL.
    * **Judge Model:** BioMedCLIP is an outdated choice. Newer, more powerful medical foundation models (e.g., BIOMEDICA) trained on far larger and more diverse datasets exist and would almost certainly provide a more reliable reward signal.
    * **Scale:** The experiments are limited to 3B and 7B models.
4.  **Marginal Improvements in Ablations:** As seen in Figure 4, the improvements from adding the $R_{VFR}$ and $R_{BPR}$ rewards are often minimal. For example, in the rightmost panel (Train on X-Ray), the test accuracy on MRI for the full model is 93.0%, while the ablation (w/o R-VFR + R-BPR) is 92.7%. A 0.3% gain is not a compelling argument for the added complexity of the method, especially given the catastrophic failure on PathVQA. The low gains on MRI data also raise questions about the judge's effectiveness on this modality.

**Questions:**

1.  Can you please provide a detailed explanation for the massive performance drop on the PathVQA dataset (Table 1) when applying REFLEX-Med, compared to the simple SFT baseline? Why does your method perform so much worse (87.8% -> 80.9% c, 79.3% -> 30.3% o)?
2.  How did you validate the quality of the saliency maps generated by the frozen BioMedCLIP judge? Can you provide quantitative or qualitative evidence that these masks are accurate, especially for the non-CXR modalities (PathVQA, CT, MRI)? Is it possible that your model is simply learning to match a *bad* set of saliency maps?
3.  The improvements in the cross-modal ablation (Figure 4) are very marginal, especially for the MRI modality (e.g., 0.3% gain in the right panel). Why do you think the gains are so small? Does this suggest the judge model is ineffective on MRI, or that the rewards themselves have limited impact?
4.  Could you clarify if the RL implementation is online or offline? The use of GRPO and sampling from the policy suggests an online setup, but this is not explicitly stated.

---

### Official Review · Reviewer_cpQL · 2025-10-31

**Soundness:** 2
**Presentation:** 1
**Contribution:** 2
**Rating:** 2
**Confidence:** 3

**Summary:**

The authors propose to use GRPO to improve the visual grounding and cross-modal provenance (without labels for these) of medical MLLMs. They evaluate their method on the medical VQA task, showing improved answer quality and faithfulness, claiming reduced hallucination and attention drift. Experiments on cross-modal (medical image modality) and zero-shot generalization are also presented.

**Strengths:**

1)Provides label label-free framework for improving visual grounding and provenance, thereby trying to reduce hallucination in large medical VLMs.
2)The framework shows good results in answer utility, cross-modal, and zero-shot performance and could be easily transferable to different backbones and settings.

**Weaknesses:**

1)The presentation of the paper could have been much better. It is hard to follow the text for various reasons. Some are listed below:
 a) The density of custom terminology is high; the core ideas are obscured by the constant use of these terms.
 b)The structure of the paper could be improved so that the reader can follow through easily without it being convoluted for no reason.
c)Some of the mathematical notations are not defined. (Ex: c, G in line 231, etc.)
d) More verbose captions for some figures (Figure 3, 6, 7) could help better understand the figure on its own.
e)Redundant description of some of the techniques/processes of the framework throughout the paper.
2)The paper claims to resist “attention-think drift” without providing any substantial evaluation.
3)The paper claims about the reasoning capabilities of their framework; they briefly analyze this in a subsection through the <think> component in their model’s responses, the details of evaluating this <think> component quantitatively, which can show the reasoning of the model, are not presented.
4)Some of the recent and relevant paper that employs GRPO for medical reasoning have been mentioned by the paper in the related work section (MedVLM-R1, MedReason), but they were not used as baselines by the paper. Justification as to why not use them as baselines was also not provided.

**Questions:**

1)The paper uses many thresholding parameters. How sensitive is the framework to the selection of these parameters? Was there any such study performed?.
2) Is there any analysis of the sensitivity of the framework to the choice and quality of the frozen judge?

---

> ### Comment · Reviewer_cpQL · 2025-11-25
> **No response posted**
>
> Since the authors have not responded to comments and questions, leaving the current rating unchanged.

---

### Official Review · Reviewer_X1rV · 2025-11-07

**Soundness:** 3
**Presentation:** 3
**Contribution:** 3
**Rating:** 6
**Confidence:** 2

**Summary:**

Motivation: medical VQA models often give plausible answers without image-grounded reasoning (“answer-right, look-wrong”), and existing explainability needs scarce rationale labels.

Proposal: REFLEX-Med, a GRPO-based RL framework with a Visual Fidelity Reward that aligns text-conditioned saliency with an anchor from the gold answer and a Bi-modal Provenance Reward that enforces text–text and image–text agreement using a frozen VLM.

Results: consistent gains over vanilla GRPO on VQA-RAD, SLAKE, and PathVQA, improved cross-modality transfer, and qualitatively tighter attention maps, with ablations showing both rewards matter.

**Strengths:**

1. The paper addresses a timely and important problem in medical vision–language modeling, namely improving answer grounding without extra process supervision.

2. VFR optimizes IoU between text-conditioned saliency masks from a frozen medical VLM, and BPR enforces text–text and image–text agreement through explicit thresholds. The pipeline is straightforward to implement and uses only answer labels, avoiding rationales, region annotations, or segmentations by deriving anchors and embeddings from the gold answers.

3. Declarativizing questions and answers into canonical statements yields a single interface for computing saliency and embeddings, which unifies close ended and open ended VQA under one policy.

4. The paper reports results across multiple medical VQA benchmarks and modalities with in-domain and out-of-domain tests, includes cross-modality transfer analyses, and provides ablations that isolate the contribution of VFR and BPR.

**Weaknesses:**

1. Faithfulness is assessed through a single frozen medical VLM judge for both saliency and embeddings. Improvements could reflect increased agreement with that judge rather than truthfulness to the image. There is no external grounding metric or human assessment of localization to break this circularity.

2. The saliency maps come from the judge without calibration. IoU between two unvalidated masks might not correlate with clinical localization quality. The paper lacks any quantitative localization benchmark or sanity checks on the saliency mechanism.

3. BLEU, ROUGE, and BERTScore are known to be poorly aligned with clinical correctness in free-form medical text. Without clinically grounded scoring or exactness checks on key entities, the reported open-ended gains may overstate clinical utility.

**Questions:**

1. Since VFR and BPR use a single frozen judge for both saliency and embeddings, the policy may align to that judge rather than the image and evaluation can become circular. How do you demonstrate that the gains reflect real grounding? Do results persist when you replace the judge with a different model after training?

2. The method uses fixed hyperparameters for the saliency quantile and thresholds. A robustness analysis to these choices would improve the contribution, as this could affect learning dynamics and reported gains.

---

> ### Comment · Reviewer_X1rV · 2025-11-26
>
> Since the authors have not responded, I will leave my assessment unchanged.

---

### Official Review · Reviewer_C48v · 2025-11-08

**Soundness:** 2
**Presentation:** 2
**Contribution:** 3
**Rating:** 2
**Confidence:** 4

**Summary:**

This paper proposed Reflex-Med as a new reinforcement learning framework for VLM. It proposed to use a CLIP model to compute (1) an image-text saliency map for fine-grained visual alignment reward, and (2) a text-text and text-image similarity score for cross-modal semantical reward. The paper has evaluated the proposed method on both in-domain and out-of-domain datasets and shows an improved overall performance against the existing baselines, demonstrating the effectiveness of the proposed method.

**Strengths:**

1. The proposed method is proven to be effective via massive experiments. The evaluation includes as many as 7 datasets from different sources and different focuses. And the proposed method demonstrated a non-trivial overall improvement against a same-size baseline model with large-scale pre-training. This is still quite impressive, considering the simplicity of the proposed method (in a positive way).

2. The proposed idea of using a frozen CLIP model for multi-modal reward is convincing. Different from a discrete accuracy reward or text-only reward, the proposed method takes the image input into consideration and measures the correlation between the model output and the image.

3. The core code is provided in the supplement.

**Weaknesses:**

First, and foremost, the paper has obviously modified the paper margin. Its bottom margin is increased, while the left and right margin is decreased. It is unclear whether the paper has gained or lost space from this modification, but I believe this is a clear violation of the conference paper requirement, which clearly describes the page margin. Given that, I think I have no choice but to reject this paper. Yet, I do have some more comments about the paper's weakness, listed below.

1. While the experimental results are impressive, the paper seems to overstate its contribution, from the reviewer's point of view. The paper claims the proposed method can help avoid hallucination and attention-think drift, and further improve the faithfulness of the reasoning and explainability.

    However, the proposed rewards only rely on the text output (modal statement $t_1$), and it is computed via a stand-alone CLIP model. This leads to two problems: **(1)** Is the CLIP model as a judge reliable? All the proposed rewards are computed as semantic similarity in the CLIP model's embedding space, which could be error-prone from the first hand. The chosen BioMedCLIP is clearly not pre-trained for fine-grained text-image alignment, making the saliency map unreliable as well. There is also no fact-check or direct chain-of-thought quality assessment, which means the claim of avoiding hallucination is questionable. **(2)** The rewards are indirect quantities in the GRPO optimization, which means optimized rewards don't guarantee a better reasoning or explainability, but just higher semantic similarity between model output and input text-image pair. *Eventually, GRPO is optimizing the model in a direction that generates output more similar to the anchor text, rather than improving the reasoning.* This could be fine in terms of improving performance, but no proof or evaluation shows that this is helpful for explainability.

2. The paper claims the proposed method is a **label-free** solution, but it is not that solid a point. Compared with all the baselines mentioned in the paper, the proposed method uses the same VQA data, where the ground-truth answer text is necessary. Of course, the proposed method does not need a fine-grained local corresponding map or heatmap, but none of the baselines or commonly used methods need these additional annotations. To better validate this point, one may want to compare with a baseline that requires such annotation.

3. The proposed method also claims it can improve the visual grounding capability. However, from the limited visual example in Figure 6, it is not obvious that the model is really capable of visual grounding. Figure 3 looks nice, but it is the saliency map for the CLIP model, rather than the actual attention of the VLM. Moreover, optimizing the VFR reward only improves the quality of the saliency map, but not the internal attention of the VLM. Providing more visual examples and reasoning results could help answer this question.

4. It might just be the problem of the reviewer, but it would be great if some of the points could be better clarified. For example, how is the saliency map computed? Also, it will be much easier to follow the paper if the frozen CLIP reward model could be clarified earlier in the paper, rather than just using a vague description.

**Questions:**

1. Can you provide some more visualization, like Figure 3 and Figure 6? The reviewer is very interested in the quality of the visual grounding for both the CLIP model and the final VLM.

2. When computing the text embedding for the model output, does it include the thinking part of the output? Or is it just about the answer part?

3. Also, the reviewer wonders what will happen when the question asks about questions related to global information, e.g., imaging modality, and how the grounding reward will be helpful in this case. Also, for a yes/no question, if the question itself is wrong, what will happen?

---

### Official Review · Reviewer_oSrf · 2025-11-10

**Soundness:** 3
**Presentation:** 3
**Contribution:** 2
**Rating:** 4
**Confidence:** 4

**Summary:**

The paper presents an RL fine-tuning framework for medical VLMs that aims to make explanations auditable without rationale labels. The method converts each (question, answer) pair into a declarative statement and evaluates two frozen signals from a medical vision-language encoder (BioMedCLIP): (1) a Visual Fidelity Reward (VFR) that grants a bonus if text-conditioned saliency for the model’s statement sufficiently overlaps (IoU) with saliency for an “anchor” statement built from the dataset ground-truth answer; (2) a Bi-modal Provenance Reward (BPR) that requires both text-text and image-text cosine similarities to exceed margins. These rewards are added to the conventional format and answer rewards and optimized with curriculum GRPO (i.e., close-ended first, then open-ended QA). Experiments are performed on multiple datasets, including both in-domain and out-of-domain evaluation.

**Strengths:**

1. The paper is well-organized and well-written.
2. The motivation is sound.
3. The experiments are conducted on six datasets.

**Weaknesses:**

1. The comparison with previous works should be improved.
- Label-Free RL has been widely explored [1][2][3][4].
- The core contributions of this work are the proposed faithful visual grounding and bi-directional cross-modal provenance rewards for RL training; however, these have already been introduced in previous studies [5][6][7].

2. The definition of “Label-free” is not clear, as the ground-truth anchors are provided during training.

3. The main claim of this paper is that the proposed method can provide auditable and faithful explanations.
- However, the paper does not include experiments to support this claim. For example, for faithfulness, the authors should evaluate the saliency maps against human-annotated ROIs.
- In addition, external benchmarks for hallucination and robustness are missing, which weakens the core anti-hallucination argument.

4. The comparison with previous works should be improved.
- Medical RL methods are not included.
- The ablation study of the curriculum setting is missing.

5. It is unclear how the saliency map 𝑆(𝐼,𝑡) is computed.

Refs:

[1] Absolute Zero: Reinforced Self-play Reasoning with Zero Data, ArXiv, 2025.

[2] Learning to Reason without External Rewards, ArXiv, 2025.

[3] Maximizing Confidence Alone Improves Reasoning, ArXiv, 2025.

[4] Unsupervised Post-Training for Multi-Modal LLM Reasoning via GRPO, ArXiv, 2025.

[5] Grounded Reinforcement Learning for Visual Reasoning, ArXiv, 2025.

[6] X-VILA: Cross-Modality Alignment for Large Language Model, ArXiv, 2024.

[7] Reinforced Cross-modal Alignment for Radiology Report Generation. ACL, 2022.

**Questions:**

See weaknesses.

---

### Official Review · Reviewer_vYv4 · 2025-11-10

**Soundness:** 3
**Presentation:** 3
**Contribution:** 3
**Rating:** 4
**Confidence:** 4

**Summary:**

This paper introduces REFLEX-Med, a reinforcement learning framework designed to provide process verifiability for unified medical VQA without requiring human annotations. The authors argue that current explainability methods, such as post-hoc saliency maps or chain-of-thought approaches that demand extensive annotations, have inherent flaws. To address this, the paper proposes two core verifiable pre-conditions: "faithful visual grounding" and "bi-directional cross-modal provenance." Specifically, the method employs a frozen medical vision-language model as a "judge" to compute two novel reward signals, a Visual Fidelity Reward (VFR) and  Bi-modal Provenance Reward (BPR). These rewards are used to fine-tune a LVLM via curriculum learning and the GRPO algorithm. Experimental results demonstrate that the proposed method outperforms baselines on several in-domain and out-of-domain medical VQA benchmarks.

**Strengths:**

1. The paper tackles a problem of critical importance and significant challenge in the medical AI domain. In high-stakes clinical scenarios, it is crucial for a model not only to provide the correct answer but also to offer a reasoning process that can be audited and trusted by physicians.
2. The paper conducts extensive experiments across multiple standard medical VQA datasets, comparing the proposed method against a range of baselines and demonstrating its superiority on several metrics.
3. The paper is generally clearly structured, and easy to follow. The authors effectively articulate the problem background, motivation, and the proposed methodology.

**Weaknesses:**

Despite its strengths, I have several major concerns regarding the novelty of the methodology, the rigor of the experimental evaluation, and the completeness of the exposition.
1. **Limited Novelty of the VFR**: The core idea of VFR, enforcing visual grounding consistency by comparing the IoU of attention maps, is not a new concept. This technique has been widely used in computer vision and multi-modal learning, for instance, in Grad-CAM [1] and its variants, as well as in conditional image-text embedding networks [2]. Furthermore, the idea of "text-visual consistency" as a regularization or reward mechanism has been explored in prior work [3-4]. Consequently, the contribution of this component feels more like an application of existing techniques rather than a fundamental innovation.
2. **Lack of Comparative Experiments**: The authors mention that the GRPO algorithm has already been applied to medical VQA tasks. However, the experimental comparison section lacks a direct comparison with existing GRPO-based medical VQA methods [5-7]. This makes it difficult for readers to accurately assess the true performance gain of the proposed method over the most relevant state-of-the-art work.
3. **Clarity Issues and Lack of Symbol Definitions**: The paper's clarity suffers in several key areas. Symbols such as $y_i$, $\pi_\theta$, $c_i$  on page 5, lines 231-232, and $r_i$ in Equation (10) appear to be used without clear prior definition, which hinders comprehension. And Equation (8) introduces a LoopTight term, but the paper fails to explain its specific function, design rationale, or how it is utilized within the algorithm.
4. **Insufficient Justification and Support for Reward Design**: The designed rewards, VFR and BPR, all use indicator functions. The paper claims this "stabilizes group-standardized advantages," but provides no theoretical derivation or experimental evidence to support this crucial assertion. Using continuous values like IoU or cosine similarity directly as rewards is a more natural choice. And the reward design introduces several key hyperparameters ($τ_{IoU}=0.5, τ_{tt}=0.8, τ_{it}=0.5$). The paper provides no basis for selecting these specific thresholds.
5. **Incompleteness of Ablation Studies**: The current ablation study only tests the scenario where VFR and BPR are removed simultaneously. This is insufficient for understanding the individual contribution of each reward component. A more comprehensive ablation study should include: 1) Experiments where only VFR is removed, and only BPR is removed. 2) An ablation on the choice of the "medical judge" model. 3) An ablation on the curriculum learning strategy.

**References**:

[1] Grad-cam: Visual Explanations from Deep Networks via Gradient-based Localization, In ICCV 2017.

[2] Conditional Image-text Embedding Networks, In CVPR 2018.

[3] Learning from Observer Gaze: Zero-shot Attention Prediction Oriented by Human-object Interaction Recognition, In CVPR 2024.

[4] Human Attention in Visual Question Answering: Do Humans and Deep Networks Look at the Same Regions? Computer Vision and Image Understanding, 2017.

[5] Medvlm-r1: Incentivizing Medical Reasoning Capability of Vision-language Models (vlms) via Reinforcement Learning, In MICCAI 2025.

[6] Medreason: Eliciting Factual Medical Reasoning Steps in llms via Knowledge Graphs, Arxiv, 2025.

[7] Med-r1: Reinforcement Learning for Generalizable Medical Reasoning in Vision-Language Models, Arxiv 2025.

**Questions:**

The questions are provided above.

---

### Meta-Review · Area_Chair_37sz · 2026-01-01

**Summary:**

The reviewers list many issues with novelty, rigor, and clarity. These issues are unresolved and the authors should address them prior to this paper's publication.

**Reviewer Concerns:**

The authors did not participate in the rebuttal period, so all concerns remain outstanding.

**Reviewer Scores:**

R1: Unchanged

R2: Unchanged

R3: Unchanged

---

### Decision · Program_Chairs · 2026-01-26

Reject